# A Global Positioning System Used to Monitor the Physical Performance of Elite Beach Handball Referees in a Spanish Championship

**DOI:** 10.3390/s24030827

**Published:** 2024-01-26

**Authors:** Alejandro Martínez-Rodríguez, Javier Sánchez-Sánchez, Jorge López-Fernández, Daniel Lara-Cobos, Juan Antonio Sánchez-Sáez

**Affiliations:** 1Department of Analytical Chemistry, Nutrition and Food Science, Alicante Institute for Health and Biomedical Research (ISABIAL), University of Alicante, 03690 Alicante, Spain; amartinezrodriguez@ua.es; 2European Institute of Exercise and Health (EIEH), University of Alicante, 03690 Alicante, Spain; 3Universidad Europea de Madrid, Faculty of Sport Sciences, 28670 Madrid, Spain; javier.sanchez2@universidadeuropea.es (J.S.-S.); jorge.lopez@universidadeuropea.es (J.L.-F.); 4Beach Handball Section, Italian Handball Federation, 00135 Roma, Italy; daniel.lara.cobos@gmail.com; 5Grupo de Investigación GDOT—Gestión Deportiva, Ocio y Tecnología, Faculty of Sport, Catholic University of Murcia, 30107 Murcia, Spain

**Keywords:** k sport morphology, team sports, sand sports, tracking system, GPS

## Abstract

Beach handball is a fully developed sporting discipline on all five continents which has attracted the attention of researchers in the last decade, resulting in a proliferation of different studies focusing on players but not on referees. The main objective of this cross-sectional research was to determine the physical demands on elite male beach handball referees in four different competitions: U18 male; U18 female; senior male; and senior female. Twelve elite federated male referees (age: 30.86 ± 8 years; body height: 175.72 ± 4.51 cm; body weight: 80.18 ± 17.99 kg; fat percentage: 20.1 ± 4.41%; national or international experience) belonging to the Technical Committee of the Royal Spanish Handball Federation were recruited for this the study. The physical demands required of referees in official matches were measured by installing a GPS device. The sampling frequency used to record their speed and distance was 15 Hz. A triaxial accelerometer (100 Hz) was used to determine their acceleration. An analysis of variance (ANOVA) between competitions with post hoc comparisons using the Bonferroni adjustment was used to compare among categories. A higher distance covered in zone 1 and speeds of 0 to 6 km-h^−1^ were recorded. Most accelerations and decelerations occurred in zones 0 and 1 (zone 0: 0 to 1 m·s^−2^; zone 1: 1 to 2 m·s^−2^). The lack of differences (*p* > 0.05) between most analysed variables suggest quite similar physical demands of the four analysed competitions. These results provide relevant information to design optimal training plans oriented to the real physical demands on referees in an official competition.

## 1. Introduction

Beach handball (BH) is a high-intensity intermittent team sport in which efforts occur every six to fifteen seconds [1,2,3]. Nonetheless, contrary to indoor handball, it has lower physical demands [1,2,4,5], probably due to the fact BH is played on sand, on a much smaller court, for a shorter duration (two sets of 10 min each, with no ties allowed), and by four players instead of seven [6]. Therefore, specific research on this sport discipline is needed.

The increasing interest in BH has led to a proliferation of several studies on BH focusing on different level categories, ages and sex, [1,7,8,9]. However, contrary to other sports such as soccer [10] or indoor handball [11], the performance of BH referees has not been analysed in any context.

Given the intermittent high-intensity nature of the game [1], it is likely referees also face high-intensity periods with short recovery breaks, which could lead to incorrect refereeing decisions due to fatigue [12]. Therefore, in line with previous studies in other sports, the identification of internal and external load demands may contribute to improving the training quality of BH referees and, therefore, enhance their performance while reducing their injury risk [13,14].

Global Positioning Systems (GPS) have been widely used to monitor the physical patterns of referees in sports [2,7,14]. It is because these devices record physical patterns such as speeds, accelerations, decelerations, or high-intensity intervals, among others, which can be used to understand the fatigue effect on referees during different competition phases [15], or to identify the most physical scenarios [16]. Therefore, they can also be used to study the physical demands of BH refereeing. But also, GPS can be used to identify whether the physical demand of refereeing varies according to the competition level (e.g., U18 to senior) and the gender of the competition (male or female), as has been described in previous studies on other team sports [17]. Accordingly, and in order to address the existing gap in the literature, the main objectives of this research were (i) to analyse the physical demands (external load) on BH elite male referees and (ii) to compare the physical demands on these referees according to the competition level (U18 and senior) and according to the gender of the competition (male and female). Consequently, referees were analysed while refereeing four different competitions (U18 male, U18 female, senior male, and senior female). It was hypothesised that the physical demands of refereeing in BH are higher in the senior competition than in the U18 competition and higher in men’s competitions than in women’s competitions.

## 2. Materials and Methods

### 2.1. Study Design

This is a descriptive and cross-sectional study.

### 2.2. Subjects

All national male referees belonging to the Technical Committee of the Royal Spanish Handball Federation and participating in the VI Spanish BH Cup held in Andalucia (southern region of Spain) were invited to participate in this research. All these referees refereed at least one match from the four analysed competitions: U18 male, U18 female, senior male, and senior female. In total, twelve male participants meeting the following inclusion criteria voluntarily agreed to participate in the study (age: 30.86 ± 8 years; body height: 175.72 ± 4.51 cm; body weight: 80.18 ± 17.99 kg; fat percentage: 20.1 ± 4.41%): (i) 5 years of experience BH refereeing and being part of the national BH circuit (Arena Handball Tour^®^); (ii) refereeing at least 5 full matches; (iii) not having physical limitations or musculoskeletal injuries; and (iv) having participated regularly in the national BH circuit (Arena Handball Tour^®^).

All the BH referees were previously informed about the objectives of the study, its methods and the risks of the research, and they provided informed written consent to be part of the research. The deontological standards recognized by the Declaration of Helsinki were considered when carrying out this research. The Ethics Committee of the University of Alicante (Spain) approved the study methodology (Expedient UA-2021-03-11). Permission was requested from the Royal Spanish Handball Federation to publish the data collected from the championship.

### 2.3. Methodology: Procedure and Instruments

Data from a total of 91 official matches out of the 125 official matches of the VI Spanish BH Cup held in Andalucia (southern region of Spain) were monitored. Data from each match set were independently registered using a Global Positioning System (GPS; WIMUPro™, RealTrack Systems, Almería, Spain) device attached to the referee’s back, according to previous studies [1,17]. The analysis was conducted considering the competition level (U18 or senior) and the gender of the league (male or female). Each referee was recorded refereeing between 6 to 18 matches.

### 2.4. GPS Variables

All measurements were registered under stable atmospheric and signal reception conditions. Information was received from up to 12 satellites (Figure 1). The selected variables were registered at a sampling frequency of 15 Hz [1,15] except for the acceleration variable, which was recorded at 100 Hz using a triaxial accelerometer (WIMUPro™, RealTrack Systems, Almería, Spain). The frequencies used ensured the validity and reliability of the GPS technology [15,18]. The recorded data were exported to SPro™ software (RealTrack Systems, Almería, Spain) for analysis.

As there were no previous studies on the selected group, the variables recorded were those used in prior research conducted on official BH matches [1], either in absolute terms or relative to the minute played: Set Duration (min); total distance (m); relative distance (m·min^−1^); High-Intensity Distance (HID; m; v > 14 km·h^−1^); Relative HID (m·min^−1^); maximum speed (MS; km·h^−1^). The distance covered was also recorded in the following speed zones (in absolute [m] or relative [m·min^−1^] terms): zone 1, 0–6 km·h^−1^; zone 2, 6–8 km·h^−1^; zone 3, 8–10 km·h^−1^; zone 4, 10–14 km·h^−1^, zone 5: 14–16 km·h^−1^; and zone 6, >16 km·h^−1^. Additionally, we also recorded, either in absolute or relative terms according to the playing time, the number of accelerations and decelerations; the distance covered while accelerating and decelerating; the peak of acceleration and deceleration; and the distance covered in five acceleration zones: Z0, 0 to 1 m·s^−2^; Z1, 1 to 2 m·s^−2^; Z2, 2 to 3 m·s^−2^; Z3, 3 to 4 m·s^−2^; and Z4, 4 to 5 m·s^−2^; as well as the distance covered in five deceleration zones (Z0, −1 to 0m·s^−2^; Z1, −2 to −1 m·s^−2^; Z2,−3 to −2 m·s^−2^; Z3, −4 to −3 m·s^−2^; and Z4, −4 to −5 m·s^−2^); and peak acceleration (pAcc.; m·s^−2^). Finally, Peak Deceleration (PDec.; m·s^−2^), the Number of High-Speed Running instances (N°HSR; v > 10 km·h^−1^), and player load (pL) were recorded both in absolute terms and relative to time [1,19,20].

### 2.5. Statistical Analyses

Data are displayed in the form of mean ± standard deviation. Before performing inferential statistical analyses, the univariate normality test (Shapiro–Wilk) and homogeneity of variances (Levene test) were conducted. A two-way analysis of variance (ANOVA; competition level × competition gender), followed by the Bonferroni post hoc adjustment test, was carried out. For all tests, a significance level of *p* < 0.05 was set. The standardized effect size was calculated for each comparison and classified as negligible (effect size (ES) < 0.2), small (ES between 0.2 and 0.6), moderate (ES between 0.6 and 1.2), or large (ES > 1.2). All statistical analyses were performed using SPSS V24.0 for Windows (SPSS Inc., Chicago, IL, USA).

## 3. Results

### Physical Demand on the BH Referees

Table 1 displays the main physical variables analysed for each competition and for the entire sample. No differences were reported when comparing the male and female leagues in the U18 category. Additionally, no differences were found in the comparison between the female U18 and the female senior competitions either. The comparison between the male U18 competition and the male senior competition showed that, on average, the senior competition lasted longer than the U18 competition (*p* = 0.006; +0.34 min; CI: 0.08 to 0.59; d = 0.37). It also elicited higher absolute accelerations (*p* = 0.006; +15.28 u.a.; CI: 4.43 to 25.89; d = 0.38), a greater absolute acceleration distance (*p* = 0.035; +13.05 m; CI: 0.93 to 25.22; d = 0.28), more absolute decelerations (*p* = 0.007; +15.03 u.a.; CI: 4.07 to 25.74; d = 0.37), and a higher player load (*p* = 0.042; +0.32 u.a.; CI: 0.01 to 6.22; d = 0.27). On the other hand, refereeing the male senior competition was associated with higher absolute accelerations (*p* = 0.008; +15.28 u.a.; CI: 3.89 to 25.53; d = 0.38), absolute decelerations (*p* = 0.015; +13.68 u.a.; CI: 2.69 to 24.55; d = 0.35), and peak accelerations (*p* = 0.016; +0.18 m·s^−2^; CI: 0.03 to 0.32; d = 0.34).

Figure 2 shows the absolute distances covered by the referees at the different speeds that define the six established zones. A greater distance covered at zone 1 speeds (0 to 6 km·h^−1^) was reported in all analysed competitions, with a longer distance covered in the U18 female competition compared to the U18 male competition (*p* < 0.05). It was observed that longer distances were covered in senior sets than in U18 sets in the other speed zones, although no significant differences were reported (*p* < 0.05). Moreover, no differences in any of the relative distance zones were found when comparing according to the competition level (U18 and senior) or the gender of the competition (male and female).

Figure 3 shows the average distances covered by the referees in the different acceleration and deceleration zones. The greatest distances during the set were covered in acceleration zones 0 and 1 (from 0 to 1 m·s^−2^ and from 1 to 2 m·s^−2^) and in deceleration zones 0 and 1 (from −1 to 0 m·s^−2^ and from −2 to −1 m·s^−2^), regardless of sex or competition. No significant differences were found when comparing between competitions and leagues, either for absolute or for relative variables (*p* < 0.05).

## 4. Discussion

This research is the first to analyse the physical demands on elite BH referees in official national competitions, considering both the competition level (U18 vs. senior) and the competition’s gender (male vs. female). However, contrary to our hypothesis, the physical demands per minute played in BH refereeing were relatively similar across the U18 national male, U18 national female, senior national male, and senior national female competitions.

The average playing time for the four studied categories ranged between 11.02 and 11.36 min. However, caution must be exercised when comparing absolute values, as the senior male competition, on average, played for significantly more time than the U18 male competition (0.34 min). Furthermore, most found differences were observed in absolute variables, but not in relative variables per played minute.

The analysis of the distance zones reveals that referees covered the greatest distances at speeds between 0 and 6 km·h^−1^, although some efforts exceeding 14 and 16 km·h^−1^ also occurred during the game. These findings, along with the high values of the acceleration and deceleration variables, suggest that BH refereeing is an intermittent activity, as reported in other team sports [5,17]. Moreover, the fact that most of the distance is covered in zone 1, with only a few meters in zones 5 and 6, coincides with data reported for BH players in training [20] and competition matches [1]. However, the referees in this study covered a greater average total distance (514.62 ± 80.54 m) per set than both female BH players (396.7 ± 158.4 m) and male BH players (445.6 ± 156.3 m) in training matches [20], as well as national BH female players in official matches [1]. This may indicate that players and referees have slightly different physical demands on them. In any case, caution is needed when comparing players’ physical performances with those of referees. Pueo et al. [3] reported higher outcomes in elite Spanish BH players than those reported in this study and in other studies [1]. Moreover, unlike referees, BH players might not play the entire set, as substitutions are permitted during the game.

Indeed, this fact might explain why certain absolute outcomes, such as total distance, appear to be higher in BH referees than those reported in previous studies. However, the relative distance covered per set in our sample was lower (46.08 m·min^−1^) compared to that reported for male players (69.7 m·min^−1^) and female players (59.8 m·min^−1^) in preparation matches and for male adolescent players (69.2 m·min^−1^), as well as female adolescent players (63.3 m·min^−1^), in official competition [21]. Accordingly, it appears that BH refereeing may have slightly different distance demands compared to those of playing BH. Further research is needed to confirm our findings and to analyse the impact of contextual variables on the physical demands on both players and referees [22]. Additionally, there is a need to compare whether the High-Intensity Distance (HID) and High-Speed Running (HSR) demands in BH refereeing are similar to those on BH players, as such data have not been previously reported, despite being identified as relevant variables in previous studies on indoor handball [23].

On the other hand, our data suggest that senior competitions and male leagues might be marginally more demanding than U18 competitions or female leagues, respectively. However, the lack of significant differences across this study’s categories does not support the necessity of implementing different training approaches based on the competition being refereed. This finding is intriguing, as previous studies in other team sports, such as soccer, have identified varying physical demands based on the competition level and player gender [17].

Regarding the acceleration and deceleration variables, we found that referees seem to reach slightly higher acceleration peaks (of 3.27 ± 0.52 m·s^−2^) compared to those reached by adolescent male and female BH players (3.08 ± 0.39 m·s^−2^/ 2.75 ± 0.33 m·s^−2^) in official matches [21], as well as compared to elite German BH players [3]. However, the deceleration peak (−3.07 ± 0.99 m·s^−2^) was slightly lower than that reported for elite German BH players (−3.34 ± 0.67 m·s^−2^) [21]. In terms of the acceleration profile, our outcomes are quite similar to those of BH players, with most the distance being covered in zones 0 and 1 (from 0 to 1 m·s^−2^ and from 1 to 2 m·s^−2^) [3]. The deceleration profile could not be compared with the referenced studies on BH as it was not reported. However, the highest outcomes were again observed in zones 0 and 1 (from −1 to 0 m·s^−2^ and from −2 to −1 m·s^−2^), which is consistent with the outcomes reported for male players in indoor handball [24]. Although the acceleration and deceleration profiles are similar for both BH and indoor handball, referees were recorded to have a higher number of accelerations in zones 1, 2 and 3 compared to BH players [3]. Additionally, they experienced more accelerations and decelerations per minute (≈29 to 31 accelerations and decelerations per minute) than indoor handball players [24] and BH players [21]. These data are important as a high frequency of accelerations and decelerations during a game may be associated with an increased risk of injury [25]. Therefore, special attention should be paid to accelerations and decelerations during training, especially considering the fact that referees, on average, cover ≈23 m acerating and another 25 m decelerating. Nonetheless, a study of worst-case scenarios could yield more insight into the most demanding periods faced by BH referees [16].

On the other hand, in contrast to the distance profile, some differences were observed in the acceleration profile of referees according to the competition level (U18 and senior) and the gender of the competition (male and female). Therefore, although the core training for referees might be quite similar regardless of the competition they are officiating, coaches should consider that senior matches tend to last slightly longer than U18 matches and may lead to a greater number of accelerations and decelerations per set. Additionally, the male league could require a higher peak acceleration.

In terms of speed and player load variables, the referees reached a maximum speed (14.28 ± 1.98 km·h^−1^) quite similar to that reported for elite BH female players per set (1st: 14.63 ± 1.71 km·h^−1^; 2nd: 14.91 ±2.30 km·h^−1^) [1]. Nonetheless, both are slower than the maximum speed reported for adolescent male players (17.3 ± 2.0 km·h^−1^) using Local Positioning System (LPS) technology [21]. Contrary to what might be expected, despite the higher number of accelerations and decelerations reported, the referees’ player load was significantly lower (≈6.1 u.a.) compared to the values reported for female players in training matches (≈13.0 u.a.) [3]. However, it was relatively close to the outcomes reported by Müller et al. [21] for official matches of the German Championship, using Inertial Measurement Units technology (6.51 ± 1.24 for female adolescent players; 8.36 ± 2.24 for male adolescent players; 7.17 ± 1.82 for male players). In addition, the difference observed in the absolute player load between the U18 and senior male categories reinforces the idea that physical demands are relatively greater in senior competitions, although the differences are not substantial.

This study also has some limitations that should be acknowledged. First, the sample size was relatively small. Additionally, referees officiated a different number of matches, and the number of matches recorded per competition was also different. Moreover, the competition lasted for 3 days, so each referee officiated several matches per day. Finally, we could not control contextual variables and other factors that might have affected the physical performance of the referees, such as recovery behaviour or diet. However, this is the first study investigating the physical performance patterns of elite referees. Furthermore, the data were recorded during real matches played by the best BH players in Spain across the four studied categories.

## 5. Conclusions

Handball referees are subjected to different physical demands than BH players, with particular differences in their acceleration and deceleration demands. Nonetheless, contrary to other team sports, the physical demands of refereeing are relatively similar for U18 and senior competitions regardless of gender, although the senior competition, and more specifically the male senior competition, seems to elicit slightly higher accelerations and player load demands of referees than the other categories.

## Figures and Tables

**Figure 1 sensors-24-00827-f001:**
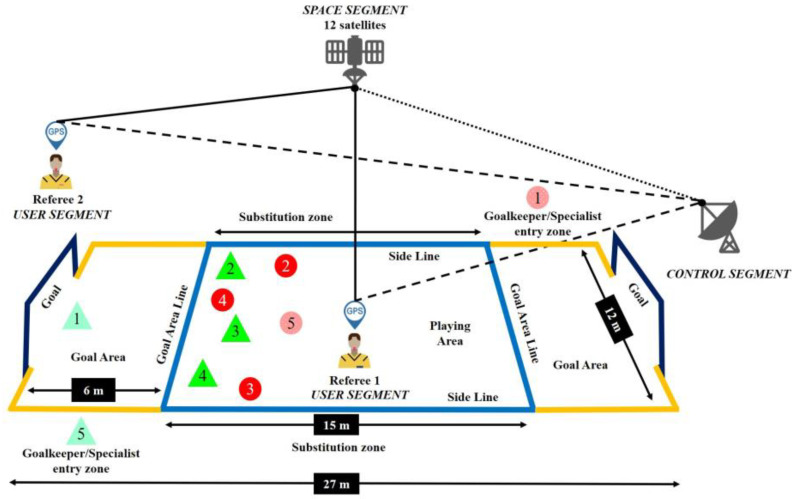
Application of the GPS system in elite BH matches.

**Figure 2 sensors-24-00827-f002:**
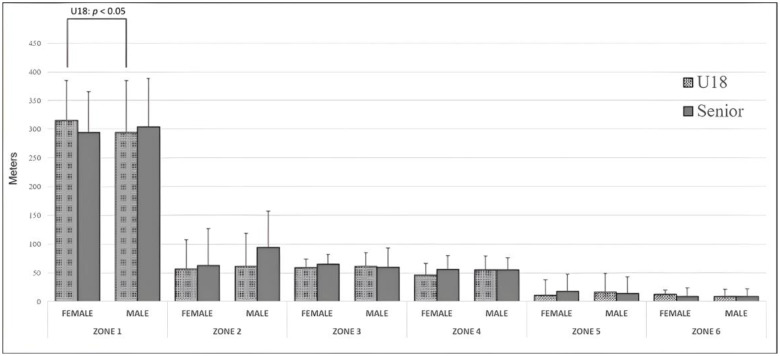
Distances covered in the speed zones according to sex and competition.

**Figure 3 sensors-24-00827-f003:**
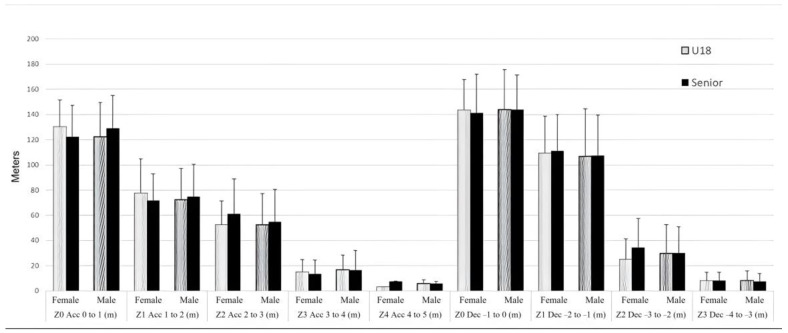
Distances covered in the acceleration and deceleration zones according to sex and competition.

**Table 1 sensors-24-00827-t001:** Physical variables for each analysed category and for the entire sample.

Variable	Whole Sample (Mean ± SD)	U18 (Mean ± SD)		Senior (Mean ± SD)	
		Male League	Female League	Male League	Female League
N of Sets	347	98	50	105	94
Duration (min)	11.18 ± 0.89	11.02 ± 0.89 ¥	11.18 ± 0.97	11.36 ± 0.95	11.12 ± 0.74
Total Distance (m)	514.62 ± 80.54	506.80 ± 90.42	513.23 ± 69.37	518.50 ± 84.42	519.16 ± 70.69
Relative Distance (m·min^−1^)	46.08 ± 6.59	46.01 ± 7.72	45.97 ± 5.52	45.67 ± 6.78	46.68 ± 5.62
High-Intensity Distance (HID; m)	49.84 ± 18.11	48.47 ± 19.94	46.83 ± 15.77	51.44 ± 18.04	51.10 ± 17.30
Relative HID (m·min^−1^)	4.46 ± 1.61	4.40 ± 1.83	4.19 ± 1.39	4.54 ± 1.57	4.59 ± 1.53
High Speed Running (M)	13.96 ± 4.94	13.63 ± 5.59	13.02 ± 4.54	14.36 ± 4.37	14.35 ± 5.01
Max Speed	14.28 ± 1.98	14.53 ± 1.79	14.45 ± 1.55	14.72 ± 1.89	14.53 ± 1.89
Accelerations (n)	347.28 ± 39.67	342.98 ± 39.64 ¥	339.72 ± 38.15	387.26 ± 41.56 ‡	343.49 ± 36,31
Accelerations (n/min)	31.11 ± 2.70	31.22 ± 2.80	30.39 ± 2.21	30.89 ± 2.63	31.23 ± 2.70
Acceleration Distance (m)	266.27 ± 44.15	259.67 ± 48.63 ¥	269.13 ± 40.29	272.71 ± 43.88	264.45 ± 40.92
Relative Acceleration Distance (m·min^−1^)	23.84 ± 3.59	23.58 ± 4.20	24.10 ± 3.28	24.02 ± 3.44	23.76 ± 3.23
Peak Acceleration	3.27 ± 0.52	3.25 ± 0.53	−2.99 ± 0.94	3.38 ± 0.60 ‡	3.21 ± 0.43
Decelerations (n)	350.93 ± 39.98	346.28 ± 40.30 ¥	344.34 ± 38.16	361.30 ± 41.84 ‡	347.63 ± 36.75
Decelerations (n/min)	31.43 ± 2.73	31.52 ± 2.88	30.80 ± 2.18	31.81 ± 2.84	31.26 ± 2.68
Deceleration Distance (m)	283.81 ± 40.72	279.80 ± 47.63	280.67 ± 34.44	284.55 ± 42.10	288.84 ± 33.87
Relative Deceleration Distance (m·min^−1^)	25.43 ± 3.31	25.41 ± 3.99	25.14 ± 2.64	25.08 ± 3.41	25.98 ± 2.66
Peak Deceleration	−3.07 ± 0.99	−3.16 ± 0.42	−3.12 ± 0.77	−3.14 ± 0.95	−3.03 ± 1.18
Player Load (u.a.)	6.06 ± 1.11	5.93 ± 1.17 ¥	5.93 ± 1.01	6.25 ± 1.19	6.06 ± 0.97
Player Load Relative (u.a·min^−1^)	0.54 ± 0.09	0.54 ± 0.10	0.53 ± 0.08	0.55 ± 0.10	0.55 ± 0.08

¥: differences between juvenile category and senior category for the given gender league. ‡: differences between male league and female league for the given category.

## Data Availability

The data presented in this study are available on request from the corresponding author. The data are not publicly available due to the presence of personal health information.

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
