# Peer review of "A Global Positioning System Used to Monitor the Physical Performance of Elite Beach Handball Referees in a Spanish Championship"

_sensors, 2024, doi:10.3390/s24030827_

Round 1
Reviewer 1 Report (New Reviewer)
Comments and Suggestions for Authors
ABSTRACT
The objective appears incomplete, since body composition and diet were also outcomes.
INTRODUCTION
Please, state the main differences between beach and standard handball.
Please, comment on previous studies analysing energy expenditure on handball referees and justify why it is important to keep searching in this field (this of course depends on whether beach differs from handball modality substantially and also on the lack of studies similar to the one proposed here).
Finally, comment on previous studies on dietary patterns on beach handball and specify the lack of studies on referees.
I see a double-fold objective here: a) energy expenditure and b) dietary patterns. Please, rephrase in accordance with this recommendation.
METHODS
How the participants were recruited. Were they (currently or previously) standard handball referees?.
Why did the authors modify the predimed questionnaire? Please, explain the changes made.
Were the referees directly interviewed, or did they answer an online questionnaire…Please, describe how data was gathered in this regard.
Minor comment: Please, rephrase this sentence as it does not appear to be grammatically correct: These measurements were crossed with the variables of competition phase, sex of the category and days of the championship
RESULTS
Information on the number of matches and their characteristics assessed in each referee should be provided (may be a supplementary file could be appropriate).
Comparisons are shown based on “first, second and third day”. I think it could be more accurate to show these differences based on “initial/final phase” of the championship.
Differences on energy expenditure taking into account sex, match level (initial/final phase) and category (Under 18/senior), should be commented.
I think is important to consider some variables that can influence energy expenditure and level of effort such as the age and the experience of the referees. Both variables are usually considered in similar studies in other sports.
The authors seem to be comparing body composition values against standard values. If so, this should be done in the discussion section. Here, they should comment whether variables such as age or sex influence body composition.
The authors state that adherence to diet is accurate, but 8 points imply “medium” adherence. Hence, this finding should be reformulated.
DICUSSION
When talking about official and training matches….are the authors referring to referees in standard handball?
Please, clearly state whether physical performance depended on the competition level (first phase vs finals), category (senior vs under age categories) and sex. There are some hints “here and there”, but there is no clear statement.
Discussion on body composition should be expanded. Why is that handball referees are not as slim as other referees, specially soccer? What about standard handball referees?
Similarly, adherence to mediterranean diet should be also discussed further. What about comparisons with the general population? What about the influence of professionalism on healthy habits of referees?
The authors understand that the practical application of this study is to recommend fitness test. I tend to disagree. Applications are related to energy expenditure and physical performance, so….what fitness dimensions should be trained/stimulated in these referees? Should they be advised to control their body weight and to improve their eating habits?
CONCLUSION
I find this section too long. The authors should be more concise.
Comments on the Quality of English LanguageMinor changes are needed. Some sentences should be rephrased.
Author Response
Dear Reviewer,
Thank you very much for reviewing our manuscript and for your insightful comments. We have exerted our utmost effort to address them.
After a careful analysis of your comments and those from Reviewer 2, we have decided to change the focus of the manuscript and concentrate exclusively on analyzing the physical demands of refereeing in the five analyzed categories: U18 male; U18 female; senior male; senior female.
Please, find attached our response to your comments

Reviewer 2 Report (New Reviewer)
Comments and Suggestions for Authors
Dear Authors,
Your submitting manuscript entitled "Global Positioning System Used to Monitor the Physical Performance of Elite Beach Handball Referees in a Spanish Championship" is valuable for this field of research. However, I recommend minor revisions to enhance clarity and coherence before proceeding.
Abstract
You should consider stating the percentage of the differences between variables in the abstract.
Introduction
Row 54 and 55. "This technology allowed us to know, for the first time, the external load needs..." Indeed, do these technologies allow us to know the external load needs? These technologies allow us to know only distance, velocity, and acceleration.
The paragraph justifying the connection between variables (distance, acceleration, etc.) and energy expenditure is missing. You should add this paragraph, including relevant literature, and connect it with Beach volleyball.
The hypothesis is missing.
Methods
Row 105 - "The kinematic and kinetic variables". You measured only kinematics variables, not kinetic ones. Kinetic variables are expressed by Newton's laws usually in the inertial system, for example, maximum strength, impulse, or impact. You have only measured movement in space and time.
Row 150 – You should consider stating Cohen's delta to express differences between two variables that had significant differences.
Discussion
The answer to the hypothesis is missing and should be stated in the last paragraph of the introduction.
Row 247 - 254: You should consider expressing the differences by using percentages.
Once these revisions are addressed, I believe your manuscript will be a solid contribution to the journal.
Comments on the Quality of English LanguageThe writing style and sentence structure are good.
Author Response
Dear Reviewer,
Thank you very much for reviewing our manuscript and for your insightful comments. We have exerted our utmost effort to address them.
After a careful analysis of your comments and those from Reviewer 1, we have decided to change the focus of the manuscript and concentrate exclusively on analyzing the physical demands of refereeing in the five analyzed categories: U18 male; U18 female; senior male; senior female.
Please, find attached our response to your comments

Round 2
Reviewer 1 Report (New Reviewer)
Comments and Suggestions for Authors
No further comments
This manuscript is a resubmission of an earlier submission. The following is a list of the peer review reports and author responses from that submission.
Round 1
Reviewer 1 Report
Comments and Suggestions for Authors
As reported by the authors, the study has some limitations:
- the sample
-body composition; the method used does not provide good estimates.
Overall, investigating a "new" population can be interesting. The use of the Mediterranean diet questionnaire is good and the study is well done.
Reviewer 2 Report
Comments and Suggestions for Authors This paper uses GPS to monitor the performance of beach handball referees at the official Spanish championships. The content of this article is relatively substantial, the structure is completed, and the English writing quality of the article is good. However, the novelty of this work is not clearly emphasized, and some parts of the paper require more explanation. The title is "GPS used to monitor the performance of beach handball referees at the official Spanish championship". However, there is not much research on GPS. It is only stated in the introduction, "These physical demands have been obtained in the last decade using the Global Positioning System (GPS), which has become one of the most widely used technologies for monitoring the physical load of referees in different team sports". What other commonly used technologies are currently used to determine the physical demands of elite beach handball referees? Are there any technical limitations in GPS technology? Did the author use new technology in the process of using the GPS system to meet the monitoring requirements of the beach handball scene and improve the measurement accuracy? The author should clearly extract the innovations of this work. Comments on the Quality of English LanguageMinor edits to English required
Reviewer 3 Report
Comments and Suggestions for Authors
Dear Authors
You have written an interesting paper focusing on the physical demands
of elite beach handball referees via GPS technology throughout the course of an official competition.
However, I don't see a point in comparing nutritional/dietary habits with the movement patterns via GPS in referees. It looks like this topic is forced into the paper and has no logical meaning. Therefore, It distorts the main point and focus of the paper. The diet has been mentioned only in 2 sentences in the discussion.
Also in this form, it does not fit well with the main scope of the Sensors journal.
Additionally, the discussion is poorly written and does not connect well to the existing literature that is done on the referees from handball or similar sports.
Therefore, I have to reject this paper in its current form.
Comments on the Quality of English Language
Moderate quality of English language.